# Chemoselective catalytic hydrodefluorination of trifluoromethylalkenes towards mono-/gem-difluoroalkenes under metal-free conditions

Jingjing Zhang[1], Jin-Dong Yang 💿 [1✉] & Jin-Pei Cheng 💿 [1,2✉]

Fluorine-containing moieties show significant effects in improving the properties of functional molecules. Consequently, efficient methods for installing them into target compounds are in great demand, especially those enabled by metal-free catalysis. Here we show a diazaphospholene-catalyzed hydrodefluorination of trifluoromethylalkenes to chemoselectively construct *gem*-difluoroalkenes and terminal monofluoroalkenes by simple adjustment of the reactant stoichiometry. This metal-free hydrodefluorination features mild reaction conditions, good group compatibility, and almost quantitative yields for both product types. Stoichiometric experiments indicated a stepwise mechanism: hydridic addition to fluoroalkenes and subsequent $\beta$-F elimination from hydrophosphination intermediates. Density functional theory calculations disclosed the origin of chemoselectivity, regioselectivity and stereoselectivity, suggesting an electron-donating effect of the alkene-terminal fluorine atom.

---

[1] Center of Basic Molecular Science, Department of Chemistry, Tsinghua University, Beijing, China. [2] State Key Laboratory of Elemento-organic Chemistry, College of Chemistry, Nankai University, Tianjin, China. ✉email: jdyang@mail.tsinghua.edu.cn; jinpei_cheng@mail.tsinghua.edu.cn

The introduction of fluorine-containing motifs is a commonly used strategy for improving the properties of target molecules in medicines, agrochemicals, and materials[1–8], because of the specific characteristics of the fluorine atom, e.g., high lipophilicity, good absorbability, and strong electron-withdrawing ability[5,6,9]. Among these important fluorine-containing motifs, *gem*-difluoroalkenes[10–13] and terminal monofluoroalkenes[14–17], deemed, respectively, as mimics of carbonyl and amide groups, have attracted much attention in the modification of bioactive molecules (Fig. 1). For example, replacement of the carbonyl group in artemisinin by a *gem*-difluoroalkene can improve its antimalarial activity (Fig. 1a)[10–12]. In some cases, the introduced *gem*-difluoroalkene moiety reverses the regioselectivity of enzyme-catalyzed hydridic reduction, overriding conventional reduction[12]. Monofluoroalkenes with high stereoselectivities are also important fragments in bioactive compounds (Fig. 1b)[18–22]. These fluoroalkenes can also serve as versatile building blocks in the construction of other fluorine-containing molecules[23–25]. Consequently, the important functions of fluoroalkenes provoked a great demand for relevant synthetic strategies[26–32].

As known, *gem*-difluoroalkenes could be conventionally constructed by functional-group interconversion, as represented by carbonyl olefination[23,33] via Julia–Kocienski,[34–37] Homer–Wadsworth–Emmons[18] and Wittig reactions (Fig. 2a)[37–39]. However, these reactions usually involve the preparation of complicated fluorinated precursors and are usually conducted under harsh conditions, e.g., in strongly basic solutions, thus leading to a quite limited substrate scope. One other good alternative could be defluorination of polyfluoroalkenes via transition-metal catalysis[40–45], photocatalysis[46–49], or the classical $S_N2'$ reactions[50–53], delivering functionalized fluoroalkenes via alkenylation[54], arylation[42], borylation[55,56], or hydrodefluorination (HDF) (Fig. 2b)[45]. Recently, Jubault, Poisson, and coworkers diastereoselectively synthesized monofluoroalkenes from the corresponding trifluoromethyl alkenes via successive dual-HDF with stoichiometric LiAlH$_4$ (Fig. 2b)[57]. However, the use of the strong hydridic LiAlH$_4$ made the reaction incompatible with

some electron-deficient groups, and this leads to undesirable over-reduction. Until now, most of the reported methods have only furnished either *gem*-difluoroalkenes or terminal monofluoroalkenes. Very recently, Wang and coworkers developed an aluminum-catalyzed tunable halodefluorination of trifluoroalkyl-substituted alkenes via fluoride ion abstraction (Fig. 2c)[58]. In their system, an arbitrary number of fluorine atoms can be selectively replaced with chlorine or bromine atoms by modification of reaction conditions. However, the reaction suffered from drawbacks concerning excess use of one of the reactants (about 4 equiv.), moderate yields, unsatisfactory stereoselectivities, long reaction time (24–48 h) and high reaction temperatures (up to 120 °C).

In this work, we described a method for metal-free catalytic activation of C–F bonds in trifluoromethylalkenes under mild conditions. *gem*-Difluoroalkenes and terminal monofluoroalkenes can be chemoselectively produced in almost quantitative yields by simple adjustment of the amount of the terminal reductant PhSiH$_3$ (Fig. 2c).

## Results and discussion

**Investigation of the reaction conditions.** Building on our previous work[59], we envisioned that diazaphospholenes of super hydricity[60–62] may provide a good chance to realize HDF of trifluoromethylalkenes via an $S_N2'$ path[63]. A preliminary attempt indicated that the reaction of α-trifluoromethyl-styrene **2a** with diazaphospholene **1** primarily gave the hydrophosphination intermediate **A**. Only handful HDF products **3a** and **1-F**[64] were obtained (Fig. 3a and Supplementary Figs. 1 and 2). However, the intermediate **A** was completely converted to **3a** and **1-F** at an elevated temperature (70 °C). This is the rare example of *β*-F elimination enabled by a non-metal neutral reductant, rather than by the well-established metal systems[65]. When a second equivalent of diazaphospholene **1** was used, the in situ-generated *gem*-difluoroalkene **3a** was further hydrodefluorinated at 70 °C to afford the dual-HDF product **4a** in an almost quantitative yield after 1 h (Fig. 3b and Supplementary Figs. 3 and 4). The excellent

**a  Bioactive molecules with *gem*-difluoroalkenes**

artemisnin
antimalarial

artemisnin analogue
**improved activity**

seletracetam
anticonvulsant

antiherpes activity

TDP-6-deoxy-L-lyxo-4-hexulose
recognized by TDP-L-rhamnose synthase
**reversal of regioselectivity**

insecticide

aminostransferferase inhibitor

**b  Bioactive molecules with terminal monofluoroalkenes**

regulator of calcium and
phosphorus homeostasis

anti-bacterial

anti-cancer

**Fig. 1 Bioactive molecules with *gem*-difluoroalkene and monofluoroalkene moieties. a** Examples of bioactive molecules with *gem*-difluoroalkene motifs. **b** Examples of bioactive molecules with terminal difluoroalkene motifs.

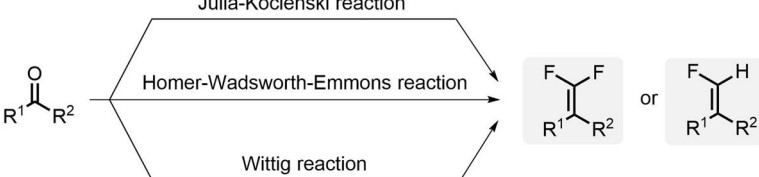

**a**  Synthesis of *gem*-difluoroalkenes and monofluoroalkenes via carbonyl olefination

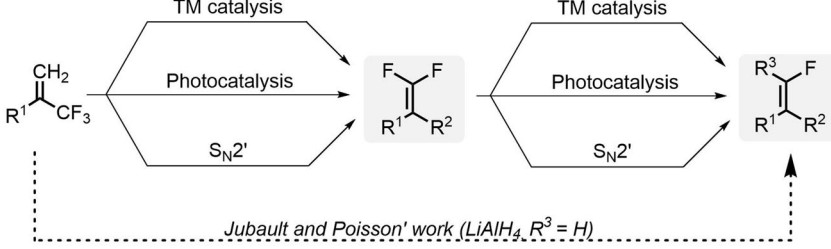

**b**  Synthesis of *gem*-difluoroalkenes and monofluoroalkenes via CF₃-substituted alkenes

**c**  Tunable defluorination reactions

*Wang's work:*

*This work:*

**Fig. 2 State of the art strategies for synthesis of *gem*-difluoroalkenes and monofluoroalkenes and this work. a** Synthesis of *gem*-difluoroalkenes and monofluoroalkenes via carbonyl olefination. **b** Synthesis of *gem*-difluoroalkenes and monofluoroalkenes via CF₃-substituted alkenes. **c** Tunable defluorination reactions.

**a**  Stoichiometric reactions of trifluoromethyl alkene **2a** with diazaphospholene **1**

**b**  Stoichiometric reactions of *gem*-difluoroalkene **3a** with diazaphospholene **1**

**c**  Regeneration of diazaphospholene **1** with PhSiH₃

**Fig. 3 Investigation of reaction conditions. a** Stoichiometric reactions of the alkene **2a** with diazaphospholene **1**. **b** Stoichiometric reactions of **3a** with **1**. **c** Regeneration of diazaphospholene **1**.

performance of diazaphospholene **1** in multiple-HDF prompted us to develop its chemoselective HDF of trifluoromethylalkenes for the synthesis of *gem*-difluoroalkenes and monofluoroalkenes. The successful regeneration of diazaphospholene **1** with fluorophilic PhSiH$_3$ via σ-bond metathesis suggested the possibility of a catalytic version of our design (Fig. 3c and Supplementary Fig. 5)[66].

**Scope of mono-HDF reactions**. As expected, mono-HDF of trifluoromethylalkenes **2** in CH$_3$CN with a 5 mol% catalyst loading and 0.33 equiv. of PhSiH$_3$ (i.e., 1 equiv. of Si–H bonds) proceeded smoothly to give *gem*-difluoroalkenes **3**. The high-polarity solvent CH$_3$CN favors polar hydride transfer. Other solvents, like toluene and THF, gave the mono-HDF products in <10% yields. The reaction showed a wide substrate scope, as shown in Fig. 4. Generally, reductions performed with either electron-rich or -deficient substrates all occurred facilely to give almost quantitative yields. Substrate **2a** furnished *gem*-difluoroalkene **3a** in 99% yield after 12 h at 70 °C. Phenyl-substituted **2b** and non-substituted **2c** were efficiently reduced at lower temperatures. The reactions of substrates with electron-donating groups such as methoxy (**2d**), methylthio (**2e**), and dimethylamino (**2f**) also worked well, but needed slightly higher reaction temperatures and longer reaction times. Substrates bearing electron-withdrawing groups (**2g–2l**) showed high reactivities and gave the corresponding *gem*-difluoroalkenes **3g–3l** in 91–99% yields. Notably, several functional groups (**2j–2l**) that are incompatible with the strong bases used in Wittig, Julia–Kocienski, and Homer–Wadsworth–Emmons reactions, or with strong nucleophiles in S$_N$2'-type reactions, are well tolerated in our system. The known reduction[67] of aryl ketones by diazaphospholenes was completely depressed by the higher electrophilicity of the trifluoromethyl group in **2j**. Substrate **2m** with a susceptible acetal moiety furnished the product **3m** quantitatively (99%) in a prolonged reaction time. The reaction was also applicable to the naphthalene analog **2n**. The excellent performance in heterocyclic systems (**3o–3r**) shows that this reaction can chemoselectively reduce the trifluoromethyl moiety while leaving other unsaturated structures intact[68,69]. For tri-substituted alkenes, only the *Z* isomer of **2s** is applicable (the *E* isomer of **2s** did not work, see SI for details), and **3s** is produced in 91% yield. This is probably because of a steric effect in the initial hydride transfer. The low efficiency of the reaction of the endocyclic alkene **2t** is probably also ascribable to a steric effect. Aliphatic trifluoromethylalkenes did not work in our conditions due to the low electrophilicity.

**Scope of dual-HDF reactions**. Because of the super-hydricity of the catalyst **1**, the produced *gem*-difluoroalkenes **2** continued to react with the hydride **1**. This explains why addition of a further 0.33 equiv. of PhSiH$_3$ gave dual-HDF products. Such a result indicates that chemoselective HDF can be achieved by simply adjusting the stoichiometry of the reactants. Accordingly, the preparation of monofluoroalkenes **4** by dual-HDF with 0.7 equiv. of PhSiH$_3$ (i.e., approximately 2 equiv. of Si–H bonds) was investigated. The results are summarized in Fig. 5. Overall, the reduction showed prominent chemoselectivity for most CF$_3$-containing substrates **2**, and monofluoroalkenes **4** were generated quantitatively with good to excellent stereoselectivities, although the slightly elevated temperature (80 °C) was necessary for several electron-rich substrates. For examples, substrates **2a–2c** gave monofluoroalkenes **4a–4c** quantitatively with good *E*/*Z* stereoselectivities. Electron-donating groups (**2d–2f**) did not significantly depress successive C–F activations, and **4d–4f** were obtained in good to excellent yields (75%–99%). Various electron-withdrawing groups (**2g–2l**) were also well tolerated (**4g–4l**, 92–99% yields). Notably, the acetal moiety of **2m** was not

sensitive to the dual-HDF conditions. Naphthalene **2n** gave **4n** in an excellent yield and with good diastereoselectivity. The heterocyclic trifluoromethylalkenes **2o–2r** were also compatible and gave products **4o–4r** in 64–99% yields with moderate to good *E*/*Z* stereoselectivities. Similarly to mono-HDF, only the *Z* isomer of **2s** showed reactivity in dual-HDF. The exocyclic trifluoromethylalkene **2t** did not react at all. Note that further increase of the amount of PhSiH$_3$ could remove the third fluoride from some substrates with electron-withdrawing groups.

**Synthetic applications**. To show the versatility of the present system, we used its potential for modifying drug molecules. Indometacin is a commonly used drug, which has significant antipyretic, anti-inflammatory, and antirheumatic activities[70]. As shown in Fig. 6, under our reaction conditions the indometacin derivative **5** is effectively transformed into either the mono-HDF product **6** in 85% yield or the dual-HDF product **7** in 30% yield, depending on the amount of the reductant PhSiH$_3$.

**Mechanistic investigations**. Density functional theory (DFT) calculations were used to gain mechanistic insights into the outstanding catalytic performance of diazaphospholene **1** in HDF. The calculations were performed at the (SMD)-M06-2X/6-311++G(2df,2p)//(SMD)-M06-2X/6-31+G(d) level of theory[71,72] with trifluoromethylalkene **2c** as the template substrate. The results are shown in Fig. 7. During the first HDF, hydride transfer from diazaphospholene **1** to **2c** proceeds via **TS1**, with a Gibbs activation barrier of 18.9 kcal mol$^{-1}$, to generate the hydrophosphination intermediate **A**, in line with our room-temperature reaction conditions for initial hydrophosphination (Fig. 3a). Exothermic *cis*-β-F elimination from intermediate **A** furnishes the mono-HDF product **3c** via **TS2**, with a 20.7 kcal mol$^{-1}$ barrier. This suggests a need for elevated temperatures.

The possible paths for the second HDF are more complicated because the hydride can be transferred to either the C1 or C2 site of **3c**, to produce intermediates **B** or **B'**, respectively (Fig. 8a). Intuitively, the C2 site is preferred, because of the strong electron-withdrawing ability of the fluorine atom. However, our experimental and DFT results uniformly led to good regioselectivity for the hydride transfer to the C1 site, which proceeds via **TS3** with an energy barrier about 15 kcal mol$^{-1}$ lower than the transfer to the C2 site via **TS3'**. This result is primarily attributed to the following two factors: (1) the stabilizing effect of the aromatic ring on the incipient benzyl carbanion during hydride addition at the C1 site, and (2) the repulsive interaction between the fluorine lone pairs of electrons and π-electrons, which makes the C1 site relatively electron deficient (Fig. 8b). Our results suggest that the alkene terminal fluorine atom has an electron-donating effect rather than the conventional electron-withdrawing effect. The NPA (natural population analysis) analysis also supported the regioselectivity in the second hydride transfer process.

The energy profile for the second HDF is given in Fig. 9. As shown, hydride transfer from **1** to **3c** via **TS3** has a higher Gibbs activation barrier (24.0 kcal mol$^{-1}$) than that of **TS1** in the first HDF (18.9 kcal mol$^{-1}$). This 5.1 kcal mol$^{-1}$ difference guarantees excellent chemoselectivity for mono- and dual-HDF. A second *cis*-β-F elimination from **B** can give two isomers of the monofluoroalkene **4c**. Based on the difference between **TS4** and **TS5** (0.9 kcal mol$^{-1}$), the *E*-isomer of the monofluoroalkene, **E-4c**, is preferentially formed. This stereoselectivity can be explained by using Newman projections of the conformers involved in *cis*-β-F elimination (Fig. 10a). The preferred conformer, which results in the *E* isomer being the major product, clearly diminishes electronic repulsion between the fluorine atom and the aromatic ring. Indeed, the stereoselectivity (*E*/*Z* = 82/18) calculated from

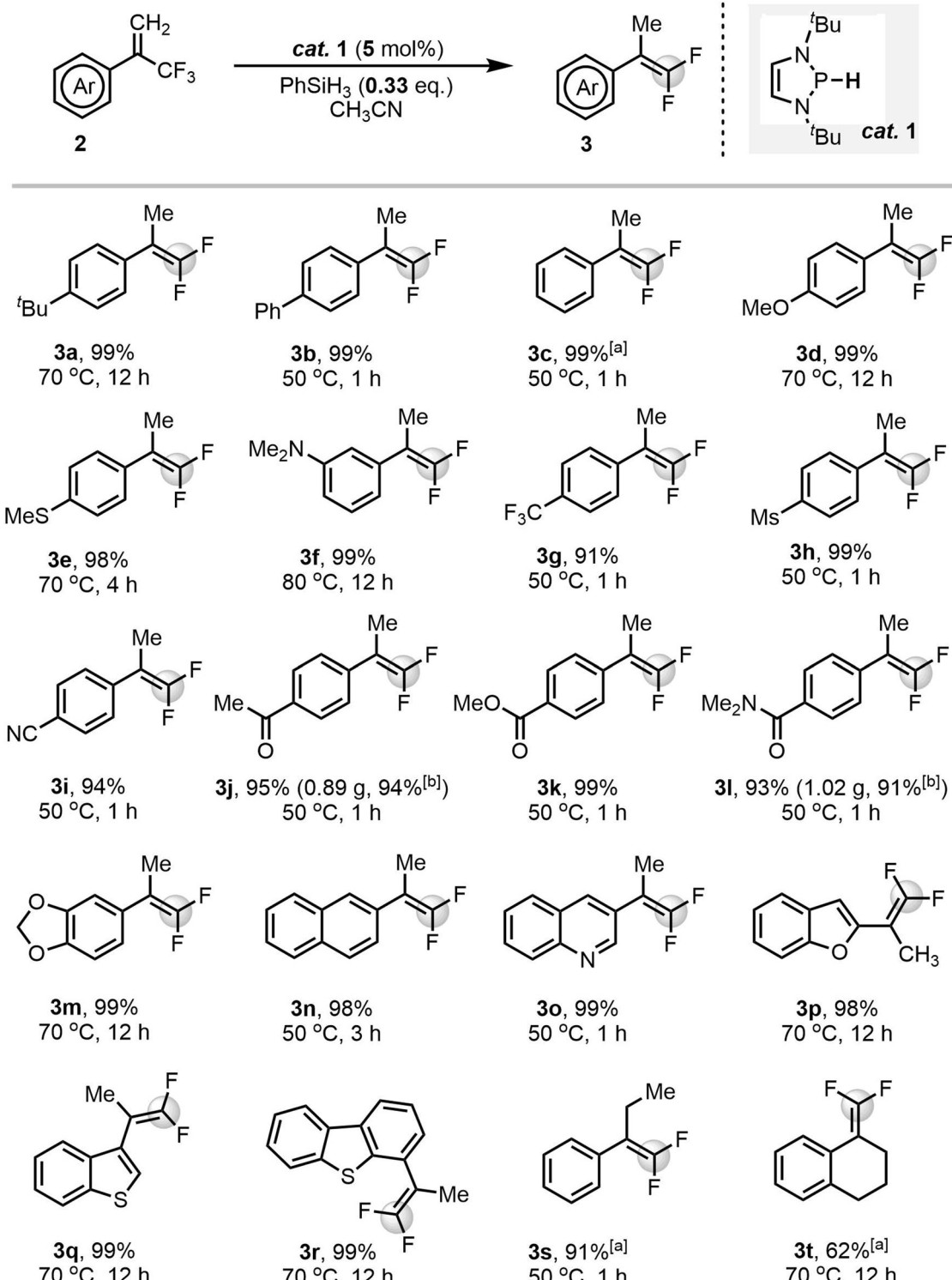

**Fig. 4 Synthesis of *gem*-difluoroalkenes by diazaphospholene-catalyzed HDF of trifluoromethylalkenes.** General reaction conditions: **2** (0.3 mmol), **1** (5 mol%), PhSiH₃ (0.33 equiv.) and CH₃CN (1 mL) were mixed in a tube under Ar. Isolated yields were given. [a] Determined by ¹⁹F NMR spectroscopy. [b] Isolated yields for gram-scale synthesis: **2j** or **2 l** (5.0 mmol), **1** (5 mol%), PhSiH₃ (0.33 equiv.), CH₃CN (5 mL), 50 °C, 3 h.

the Gibbs activation energies for both β-F elimination steps is in good agreement with the experimental result (*E/Z* = 85/15).

A plausible mechanism for this HDF process is outlined in Fig. 10b. First, hydride transfer from the catalyst **1** to trifluoromethylalkenes **2** furnishes the hydrophosphination intermediate **A**. Subsequent β-F elimination gives the mono-HDF products

**3** and **1-F**. After complete consumption of **2**, the excess PhSiH₃ regenerates catalyst **1**, which renders a second HDF to yield monofluoroalkenes **4**.

In summary, we have developed a method for diazaphospholene-catalyzed chemoselective C–F bond activation of trifluoromethylalkenes, which enables the convenient construction

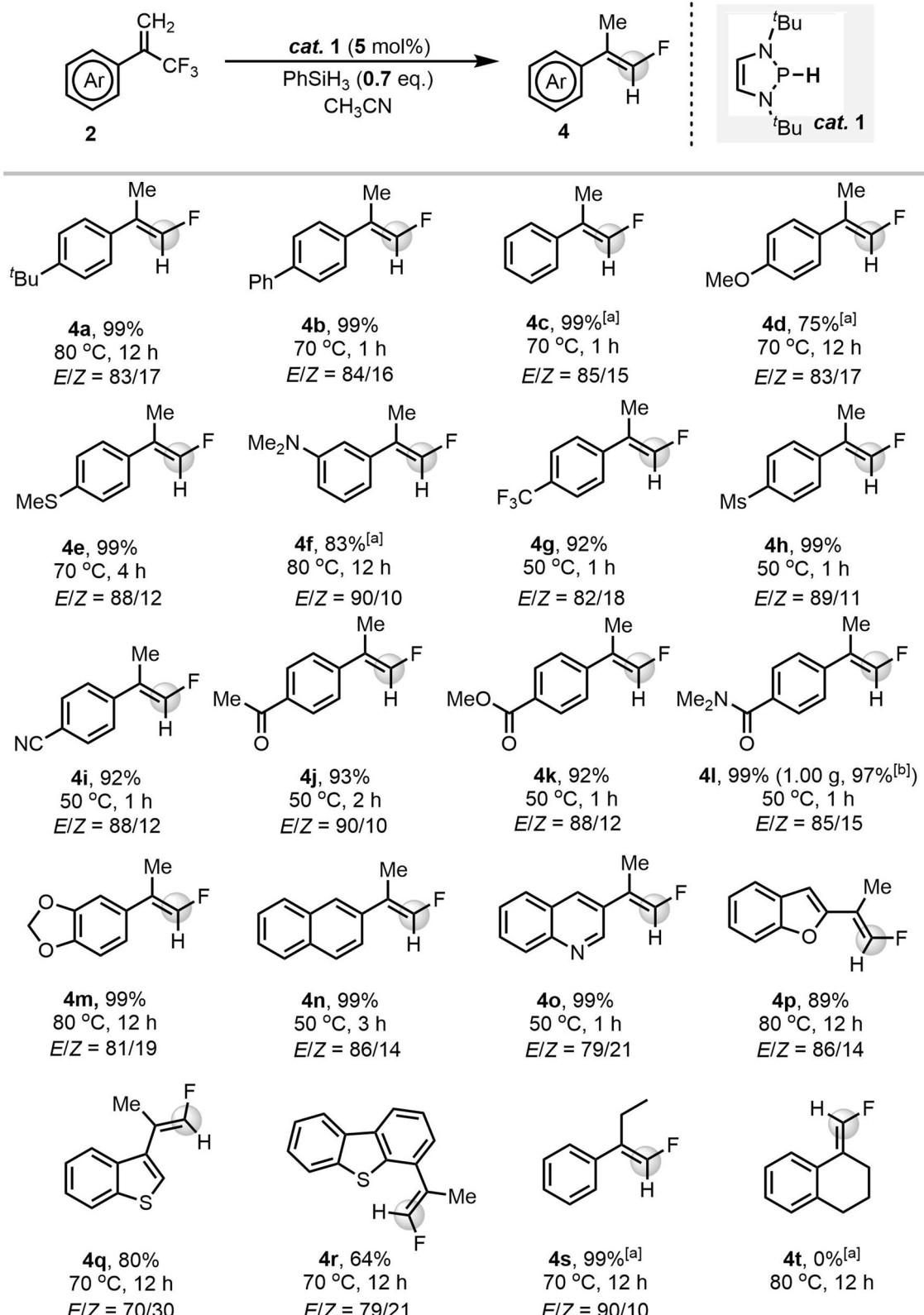

**Fig. 5 Synthesis of terminal monofluoroalkenes by diazaphospholene-catalyzed HDF of trifluoromethylalkenes.** General reaction conditions: **2** (0.3 mmol), **1** (5 mol%), PhSiH$_3$ (0.7 equiv.), and CH$_3$CN (1 mL) were mixed in a tube under Ar. Isolated yields were given. The *E/Z* ratios were determined by $^{19}$F NMR spectroscopy. [a] Determined by $^{19}$F NMR spectroscopy. [b] Isolated yield for gram-scale synthesis: **2l** (5.0 mmol), **1** (5 mol%), PhSiH$_3$ (0.7 equiv.), CH$_3$CN (5 mL), 50 °C, 3 h.

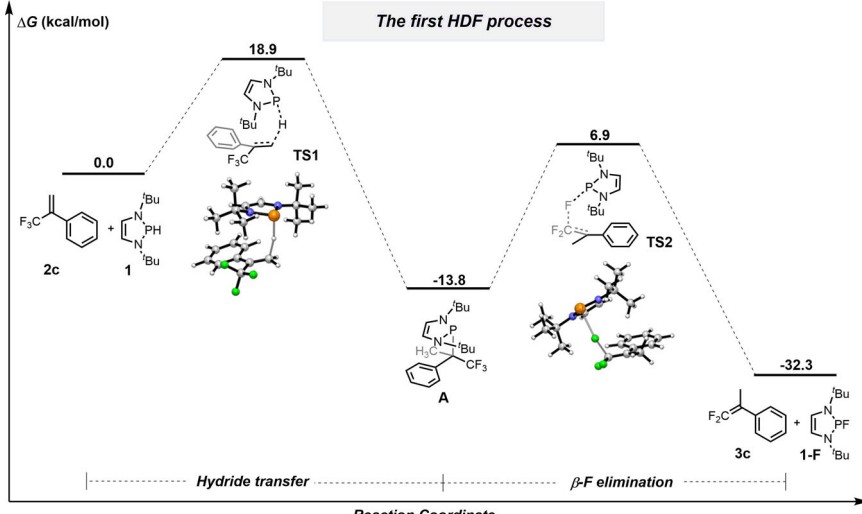

**Fig. 6 Modification of the drug molecule.** *E/Z* ratios were determined by ¹⁹F NMR spectroscopy.

**Fig. 7 Mechanistic investigations for the first HDF process by DFT calculations.** Energy profiles for mono-HDF of **2c** by **1** in acetonitrile calculated at the (SMD)-M06-2X/6-311 + +G(2df,2p)//(SMD)-M06-2X/6-31 + G(d) level of theory. All energies are in kcal mol⁻¹.

**a** Regioselectivity for the hydride transfer from 1 to 3c (in kcal mol⁻¹)

**b** Electrostatic property of the alkene-terminal fluorine atom

**Fig. 8 Rationalization of the regioselectivity of hydride transfer from diazaphospholene 1 to 3c. a** Regioselectivity for the hydride transfer from **1** to **3c**. **b** Electrostatic property of the alkene-terminal fluorine atom.

of *gem*-difluoroalkenes and terminal monofluoroalkenes under metal-free conditions with PhSiH₃ as the terminal reductant. NMR spectroscopic studies showed a hydrophosphination intermediate, which subsequently underwent *β*-F elimination at elevated temperatures. This metal-free strategy is applicable to a broad range of trifluoromethylalkenes. It shows good functional group tolerance and gives almost quantitative yields of both mono- and di-hydrodefluorinated products. DFT calculations suggested that the good chemoselectivity between mono- and dual-HDF stems from differences in the substrate electrophilicities, and the regioselectivity for hydride transfer to *gem*-difluoroalkenes is partly attributed to the electron-donating ability of the alkene terminal fluorine atoms. Other diazaphospholene-catalyzed HDF reactions are currently being investigated in our laboratory.

## Methods

**General information**. Catalyst 1 has been synthesized and characterized in our previous work[59,73,74]. Trifluoromethyl alkenes 2 were synthesized according to references (see Supplementary Information for details). Other reagents and solvent were purchased from J&K or TCI Chemicals and used without further purification unless specified otherwise. Acetonitrile was purchased from J&K Chemical (99.9%, Extra dry, water <10 ppm, J&K seal) and degassed and distilled by standard methods. Reaction temperature refers to the temperature of an aluminum heating block or a silicon oil bath, which was controlled by an electronic temperature modulator from IKA.

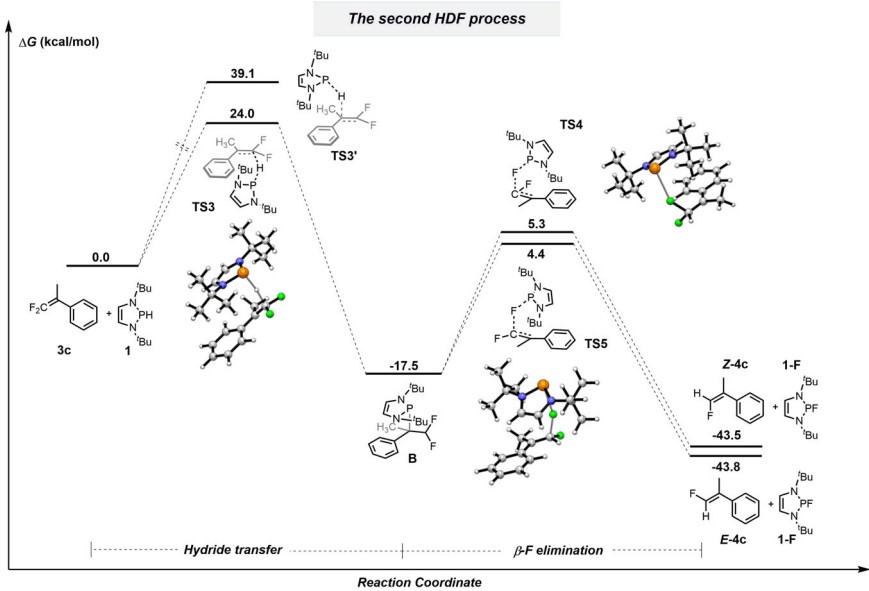

**Fig. 9 Mechanistic investigations for the second HDF process by DFT calculations.** Energy profiles for HDF of **3c** by **1** in acetonitrile calculated at the (SMD)-M06-2X/6-311 + +G(2df,2p)//(SMD)-M06-2X/6-31 + G(d) level of theory. All energies are in kcal mol$^{-1}$.

**Fig. 10 The origin of the Z/E selectivity and proposed mechanism. a** Newman projections of the intermediate **B** involved in *cis-β*-F elimination. **b** Proposed reaction mechanism.

**Reactions**. All hydrodefluorination reactions were carried out in dry glass wares under an argon atmosphere using Schlenk technique throughout the reaction procedures.

**Analytics**. $^1$H and $^{13}$C NMR, $^{19}$F NMR spectra were recorded in CDCl$_3$ ($\delta = 7.26$ for $^1$H NMR, $\delta = 77.16$ ppm for $^{13}$C NMR) on 400 MHz NMR instrument at Center of Basic Molecular Science (CBMS) of Tsinghua University.

**DFT calculations**. Geometry optimizations and frequency computations were performed using Gaussian 09[75] at the M06-2X[71,76]/6-31 + G(d) level of theory, in conjunction with the SMD[72] model to account for the solvation effect of acetonitrile. To obtain more accurate electronic energies, single point energy calculations were performed at the SMD-M06-2X/[6-311 + +G(2df, 2p)] level with the SMD-M06-2X/[6-31 + G(d)] structures.

## Data availability

The authors declare that all the data supporting the findings of this work are available within the article and its Supplementary Information files.

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

## Acknowledgements

We are grateful for the financial support from National Natural Science Foundation of China (Nos. 21973052, 21933008, 91745101) and Tsinghua University Initiative Scientific Research Program (No. 20181080083).

## Author contributions

J.-P.C. and J.-D.Y. conceived and supervised the project. The synthetic experiments and characterizations were carried out by J.Z. All authors discussed the results and commented on the manuscript.

## Competing interests

The authors declare no competing interests.
