## [Peer Review File · Nature Communications]

REVIEWER COMMENTS

Reviewer #1 (Remarks to the Author):

In this work, Cheng and Yang and Zhang carried out a chemoselective mono/ bis defluorination of vinyl CF₃ groups to make fluoro-alkenes. They used diazaphospholenes as catalysts. Diazaphospholenes are a recent development as hydride transfer catalysts, and this is a new application for diazaphospholenes. The products are useful, and I think this work is a nice addition to the field of diazaphospholene catalysis. The substrate scope appears to be limited to aryl containing compounds, though heteroaryl compounds are tolerated. The substrate scope is further limited by the fact this transformation is really only pertinent to CF₃ alkenes. The main strength is this is the first reported catalytic reaction where a diazaphospholene fluoride is the propagating species.

I have numerous comments regarding the manuscript and the supporting information, which I think need to be addressed before publication. My main scientific complaints are that the double defluorination is not as well developed as the mono defluorination, and I have specific concerns about the presented data that I will address in more detail as I address my concerns line by line. I have strong concerns that almost every substrate for both reactions has a 99% yield, after a purification step. I don't think these yields are warranted based on the cleanliness of the NMRs that are exhibited, especially in the double defluorination reaction, where almost all products are quite impure. The reactions were also run primarily on a relatively small 0.3 mmol scale, and specific masses for each recovered product were not given, further making 99% yields difficult to accept. Certain analytical data that is required for complete characterization is also missing. In many cases, crucial integrals in the NMR spectra are not shown. The double defluorination compounds are particularly missing integrals expressing the reported ratios, and in some cases (I elaborate more on this later), I think the assignment of the minor isomer is in jeopardy. I think a more thorough job of the SI needs to be undertaken, including isolated yields and proper characterization of new compounds, to adequately convince the readership of the identity and homogeneity of the product compounds.

More specific comments are given line by line:

Line 71.. An insufficient number of citations for precedent in diazaphospholene chemistry are given. While the authors have pioneered the use of **1** and related compounds in radical chemistry, the following points that based on the use of **1** in polar chemistry are also relevant:

1-F is a known compound and Gudat's preparation should be referenced (Chem Eur J, 2000). Use of PhSiH₃ as a reductant with diazaphospholene systems was previously shown by Kinjo for CO₂ reduction (ACIE 2016), and Cramer for his recent work on reduction of α,β unsaturated acids (Chem Cat Chem 2020), and these works should be referenced. Speed's recent work with reactions of **1** and the dimer of **1** for dehalogenation/ phosphination should be cited (EuJOC 2020).

Acetonitrile was the only solvent mentioned. While acetonitrile has been a privileged solvent in diazaphospholene chemistry, were other solvents that were used to great effect in diazaphospholene chemistry by Gudat, Kinjo, Cramer, and Speed such as diethyl ether or toluene, or THF attempted?

All yields are stated to be determined by ¹⁹F spectroscopy and are unreasonably high (99%). If complete HDF occurred, that would give products that would not be seen in the ¹⁹F spectrum. Isolated yields should be reported. Product isolation was done in the SI, yet recovered masses are not reported, so in the end I am confused if the 99% yield is isolated or NMR yield.

Line 97 It would be worth noting **3j** contains a ketone, which Kinjo has shown is hydroborated, via a hydrophosphination, followed by sigma bond metathesis in acetonitrile (ie reference 61). It is noteworthy this does not deactivate the catalyst by hydrophosphination.

Product **3o** contains quinoline. Are pyridines tolerated? Speed (OM 2018) showed **1** reacted with

pyridines via hydrophosphination at ambient temperature, while Kinjo (JACS 2018) also showed pyridine reduction by diazaphospholenes. These could be expected to deactivate the catalyst.

Line 104 What is meant by only Z isomer is applicable for 3s? Did only the Z isomer react, or was only the Z isomer tried? The E isomer might have the aryl twisted out of conjugation. This is alluded to in the SI, but complete characterization data for the substrate or a detailed procedure for this substrate is not required.

In Scheme 4, it is clear that only aryl-containing substrates are shown, yet this is not addressed in the text. Can CF₃ alkenes bearing alkyl substrates work in this reaction?

129 – Complete HDF is referenced, but not shown. Does this mean removal of the third fluoride?

Scheme 5... are the increased temperatures in several examples necessary for the second hydrodefluorination? This is supported by the later computations, but should be addressed.

Scheme 7. The computed barriers are reasonable, and structures are intuitive, though I do not have the qualifications to assess the appropriateness of the functionals/ level of theory that were used.

A number of questions arise from the chemistry involving the removal of 2 F atoms. In general, this section has the feel of being less well developed than the first section, and raises more questions:

Scheme 8. Scheme 8b, showing inverted polarization is a surprising result, which flies in the face of chemical intuition, and also the classic behavior of nucleophile addition to difluoroalkenes. Can this be supported using a computational analysis of the HOMO and LUMO?

The mechanistic discussion of the double dehydrofluorination is less well developed than the removal of 1 F.

Scheme 9... the energy profile implies B can be isolated.

Has this been attempted? Isolation of B, perhaps by deliberately introducing a difluoroalkene starting material and 1 would lend experimental evidence to the proposed mechanism, which relies on the surprising inverted polarization above, and this should be attempted.

A 31.1 kcal/mol delta G from B to TS5 is calculated. This seems a little high for the reaction proceeding at 80 degrees in the time given. Again this is difficult to reconcile with the experimental data.

Scheme 10.... Is the steric demand of F high enough to account for the observed selectivity? Were dipole minimization effects considered? I am a little suspicious, since two of the largest aryl groups screened, 4q and 4r have some of the poorest selectivity shown in this reaction. As I mention later, I am not convinced by the NMR data provided that the minor products are the other isomer. If the reaction is operative in other solvents, would different E/Z ratios be obtained?

Scheme 11... the structure of 4 is drawn incorrectly. The bonds are crossed, and the Z rather than E isomer is depicted.

Line 204 – The synthesis of 1 should also reference Gudat's pioneering work.

The supporting information appears to be lacking several key pieces of data required to support the characterization and assignment of the observed isomer ratios of the double defluorination products.

Line 43: The boiling point of acetonitrile is 82 degrees. The temperatures reported are of the block or bath. Was it ascertained by internal thermocouple that the reaction temperature equilibrated to that of the block or bath?

Figure S1: I am surprised that 3J PF coupling is not observed in intermediate A. Can the authors

comment? In addition, an expansion of the signal for 3a in acetonitrile (which has 2 F atoms in separate environments could be warranted).

Spectra S1 and S2 are not integrated. Is quantitative integration with these nuclei possible? The ratio of A to 1-F in figure S1 seems much larger than the ratio of A to 1F in figure S2.

Figure S4. This figure appears to be a ³¹P spectrum, but is labelled as a ¹⁹F spectrum. Can the authors assign the broad peak at 190 ppm? This seems consistent with 1-Br, or another halogen other than F, which does not seem consistent with the reaction mixture.

Line 82. While a reference was included, it would be helpful for the readership to have a scheme showing the procedure.

Line 200... should read "due to the low boiling point of the products". It should be specified what pressure was used, rather than "not to high"

I don't think the 99% yields are reasonable for the reaction scale/ multistep purifications. No MS or elemental analysis data is reported. Many of the reported NMR spectra have significant impurities. Several of the product compounds listed on page S10 and S11 are new, and should have at least HRMS data in addition to the NMR data reported. Compound 6 (line 312, page 11) does have such data.

Line 317- Complete dehydrofluorination is a bit confusing, since one F remains.... Perhaps "conversion to desired product"

Line 313 onward concerns the monofluoroalkenes, formed by removal of 2 F atoms. The yields are not at all realistic given the low NMR purities of these compounds. In addition, the integrals used to determine the reported isomer ratios are not shown on the images of the spectra.

Line 393- tabulated data for compound 4q is missing.

Line 401-data for Z isomer in ¹⁹F NMR is not shown in tabulated form. I have my doubts about the assignment of the minor isomer in this case, given the dramatically different appearance of the minor signals in the NMR spectra (addressed later)

These comments address the images of the NMR spectra:

Line 414- Integrals are missing from ¹H NMR spectrum.

Line 424- The ¹H NMR spectrum contains aliphatic impurities, and integrals are not shown.

Line 428- Integrals are missing from ¹H NMR spectrum.

Line 433 Integrals are missing from ¹H NMR spectrum.

Line 448 Integrals are missing from ¹H NMR spectrum.

Line 461- Integrals are missing from ¹H NMR spectrum.

A number of the product NMRs have extra peaks

For example line 506 ¹⁹F spectrum for 3i

Double dehydrofluorination products:

Line 565 onward. The products are messier. How was the configuration determined?

The integrals that were used to select the geometric ratios were not shown.

Products such as 4g (line 574) are quite impure.

Line 608- Why is there such a dramatic shift difference in the ¹⁹F spectrum between the "isomers". Are these in fact isomers?

Reviewer #2 (Remarks to the Author):

In the manuscript, a novel process involving trifluoromethyl-substituted styrenes is described. The reaction proceeds via new type of mechanistic cycle base on a catalyst bearing the phosphorus-hydrogen bond. Importantly, no transition metal catalyst is needed. A key feature of the manuscript is the addition of the P-H bond across the C,C double bond. The proposed mechanistic cycle is supported by quantum chemical calculations. It should be noted that this reaction mechanism stands in sharp contrast to many typical reactions of trifluoromethyl-substituted styrenes. The results are nicely presented. Overall, given the novelty of the results, I would suggest acceptance of the manuscript.

Additional comments.

1. For all solid products, melting points must be given (even for known compounds).
2. For new compounds, either HRMS or combustion analysis must be given (e.g. 3f, 3q, 3r, and many others).
3. In the procedures: "...and concentrated under vacuum (Note: due to the low boil of products, the vacuum should not be too high)" This would be difficult to reproduce. Some hint about the vacuum would be helpful.

Reviewer #3 (Remarks to the Author):

This manuscript by Yang, Cheng and co-workers presents a very important work regarding the metal-free catalytic hydrodefluorination of trifluoromethylalkenes.

The authors first examined the stoichiometric reaction of diazaphospholene [1], with α -trifluoromethyl-styrene [2a]. This gives gem-difluoroalkene [3a] and fluorinated diazaphospholene [1-F], concomitant with the hydrophosphination intermediate [A]. Subsequently, the authors confirmed the formation of [1-F] from [2a] by heating the reaction mixture. This discovery is very important. In the reported diazaphospholene chemistry, a beta-elimination process has never been achieved, which hampers the development of transition metals-like catalytic reactions. By employing PhSiH₃ as a reducing reagent, reproduction of [1] from [1-F] is successfully achieved, which allows establishing the catalytic hydrodefluorination cycle. Moreover, by simply adjusting the reactant stoichiometry, gem-difluoroalkenes and terminal monofluoroalkenes are selectively obtained. The scope of substrates is nicely examined, the transformation of Indometacin derivatives has successfully been achieved with the authors' system, and the computational studies provide the reasonable reaction pathways, in good agreement with the experimental observation.

Overall, the study is well done both experimentally and theoretically, I do not see any major technical issues. The discovery of the beta-elimination process involved in this catalytic reaction is very significant and highly original. The present work should get a wide readership and I recommend the acceptance for publication after the very minor revision on the points indicated below:

- (i) While the proposed mechanism sounds reasonable, have the authors considered a process involving radical species? Have the authors tried the reaction under the light-shielded conditions?
- (ii) By increasing the amount of PhSiH₃, is it possible to complete HDF of compound 4?
- (iii) When F atoms are replaced with other elements such as Cl or Br, do the similar reactions still proceed? Any comments on the importance of the F atoms from a viewpoint of the mechanism ~ energy would be highly appreciated.
- (iv) Regarding the recent development of diazaphospholene catalysis, neither the recent reviews (for instance, Chem. Soc. Rev., 2020, 49, 8335) nor original reports therein have been cited at all. Those should be cited properly.

Point-by-point Responses to the Reviewers' Comments (NCOMMS-21-04261)

REVIEWER COMMENTS

Reviewer #1 (Remarks to the Author):

In this work, Cheng and Yang and Zhang carried out a chemoselective mono/bis defluorination of vinyl CF_3 groups to make fluoro-alkenes. They used diazaphospholenes as catalysts. Diazaphospholenes are a recent development as hydride transfer catalysts, and this is a new application for diazaphospholenes. The products are useful, and I think this work is a nice addition to the field of diazaphospholene catalysis. The substrate scope appears to be limited to aryl containing compounds, though heteroaryl compounds are tolerated. The substrate scope is further limited by the fact this transformation is really only pertinent to CF_3 alkenes. The main strength is this is the first reported catalytic reaction where a diazaphospholene fluoride is the propagating species.

I have numerous comments regarding the manuscript and the supporting information, which I think need to be addressed before publication. My main scientific complaints are that the double defluorination is not as well developed as the mono defluorination, and I have specific concerns about the presented data that I will address in more detail as I address my concerns line by line. I have strong concerns that almost every substrate for both reactions has a 99% yield, after a purification step. I don't think these yields are warranted based on the cleanliness of the NMRs that are exhibited, especially in the double defluorination reaction, where almost all products are quite impure. The reactions were also run primarily on a relatively small 0.3 mmol scale, and specific masses for each recovered product were not given, further making 99% yields difficult to accept. Certain analytical data that is required for complete characterization is also missing. In many cases, crucial integrals in the NMR spectra are not shown. The double defluorination compounds are particularly missing integrals expressing the reported ratios, and in some cases (I elaborate more on this later), I think the assignment of the minor isomer is in jeopardy. I think a more thorough job of the SI needs to be undertaken, including isolated yields and proper characterization of new compounds, to adequately convince the readership of the identity and homogeneity of the product compounds.

Reply: Thank you very much for your recommendation as well as your very professional and critical comments! As kindly advised, we re-examined the defluorination experiments for most of the substrates. The results are consistent well with those previously presented. We also carried out gram-scale preparation for **3j**, **3i**, and **4i**, which gave excellent isolated yields (91-96%) as well. The product characterization was reinforced, including but not limited to the further purification of the products, re-calculation of product yields, integration of NMR spectra, addition of MS and melting point data to SI, and rearrangement of SI. For details, please see the point-by-point response below and SI.

More specific comments are given line by line:

Line 71. An insufficient number of citations for precedent in diazaphospholene chemistry are given. While the authors have pioneered the use of **1** and related compounds in

radical chemistry, the following points that based on the use of 1 in polar chemistry are also relevant:

1-F is a known compound and Gudat's preparation should be referenced (Chem Eur J, 2000). Use of PhSiH_3 as a reductant with diazaphospholene systems was previously shown by Kinjo for CO_2 reduction (ACIE 2016), and Cramer for his recent work on reduction of α,β unsaturated acids (Chem Cat Chem 2020), and these works should be referenced. Speed's recent work with reactions of 1 and the dimer of 1 for dehalogenation/phosphination should be cited (EuJOC 2020).

Reply: Thanks. We cited these relevant references as refs. 60-66.

Acetonitrile was the only solvent mentioned. While acetonitrile has been a privileged solvent in diazaphospholene chemistry, were other solvents that were used to great effect in diazaphospholene chemistry by Gudat, Kinjo, Cramer, and Speed such as diethyl ether or toluene, or THF attempted?

Reply: Acetonitrile, a frequently-used solvent for polar reductions, was chosen based on our another (unpublished) work. Diethyl ether was found unsuitable for diazaphospholene because it causes decomposition. Toluene and THF were also examined but gave the mono-HDF products in < 10% yields. We added these results in the revised text: "*Other solvents, like toluene and THF, gave the mono-HDF products in < 10% yields.*".

All yields are stated to be determined by ^{19}F spectroscopy and are unreasonably high (99%). If complete HDF occurred, that would give products that would not be seen in the ^{19}F spectrum. Isolated yields should be reported. Product isolation was done in the SI, yet recovered masses are not reported, so in the end I am confused if the 99% yield is isolated or NMR yield.

Reply: We are sorry for causing your confusion! As specified in the footnotes of Scheme 4 and 5 in the text, most of the product yields were given as the isolated yield. Only the 3 cases for mono-HDF and 4 cases for dual-HDF were characterized in-situ by the ^{19}F and ^1H NMR spectra. The ^1H spectra clearly showed that when < 2 equiv. of the effective reductant (< 0.7 equiv. of PhSiH_3) was used, none of complete HDF occurred. We added recovered masses for all products to the experiment part in SI.

Line 97 It would be worth noting 3j contains a ketone, which Kinjo has shown is hydroborated, via a hydrophosphination, followed by sigma bond metathesis in acetonitrile (ie reference 61). It is noteworthy this does not deactivate the catalyst by hydrophosphination.

Reply: As reported in ref. 61 (now ref. 67), the hydrophosphination of ketone occurred slowly (6 h for benzophenone) at 90 °C, which is unable to compete with HDF (at 50 °C and in 1-2 h). When 1-2 equiv. of the effective reductant (0.33-0.7 equiv. of PhSiH_3) was used in our system, HDF occurred preferentially.

Product 3o contains quinoline. Are pyridines tolerated? Speed (OM 2018) showed 1 reacted with pyridines via hydrophosphination at ambient temperature, while Kinjo (JACS 2018) also showed pyridine reduction by diazaphospholenes. These could be expected to

deactivate the catalyst.

Reply: As you might expect, pyridines are not tolerated due to hydrosilylation of the pyridine skeleton.

Line104 What is meant by only Z isomer is applicable for 3s? Did only the Z isomer react, or was only the Z isomer tried? The E isomer might have the aryl twisted out of conjugation. This is alluded to in the SI, but complete characterization data for the substrate or a detailed procedure for this substrate is not required.

Reply: The **2s** was used as a mixture of the isomers (*E/Z* 1/1). Only the Z isomer showed a satisfactory reactivity. The *E* isomer didn't react at all. To avoid potential misunderstanding, we added a statement in the text: "The *E* isomer did not work."

In Scheme 4, it is clear that only aryl-containing substrates are shown, yet this is not addressed in the text. Can CF₃ alkenes bearing alkyl substrates work in this reaction?

Reply: Alkyl CF₃-substrates cannot work due to the low electrophilicity. This was specified in the text: "*Aliphatic trifluoromethylalkenes cannot work in our conditions due to the low electrophilicity.*"

129 – Complete HDF is referenced, but not shown. Does this mean removal of the third fluoride?

Reply: Yes. The third fluoride could be removed, but in a relatively low yield, if more than 2 equiv. of the effective reductant was used. We revised this description in the text: "*Note that further increase of the amount of PhSiH₃ could remove the third fluoride from some substrates with electron-withdrawing groups.*"

Scheme 5... are the increased temperatures in several examples necessary for the second hydrodefluorination? This is supported by the later computations, but should be addressed.

Reply: The mono- and dual-HDF generally occurred in a comparable temperature range (50–80 °C) with the later at slightly higher temperatures. This is consistent with their similar overall reaction barriers (20.7 kcal/mol vs 24.0 kcal/mol, please see the following DFT calculation part for details). In the dual-HDF, the elevated temperature (80 °C) was necessary for several electron-rich substrates bearing ^tBu, NMe₂ and acetal groups. We have mentioned this in revised the text: "... *although the slightly elevated temperature (80 °C) was necessary for several electron-rich substrates.*"

Scheme 7. The computed barriers are reasonable, and structures are intuitive, though I do not have the qualifications to assess the appropriateness of the functionals/ level of theory that were used.

Reply: Theoretical calculations were performed at the (SMD)-M06-2X/6-311++G(2df,2p)//(SMD)-M06-2X/6-31s+G(d) level of theory, which is one of the most frequently-used basis set for evaluation of the energy profiles for main-group elements.

A number of questions arise from the chemistry involving the removal of 2 F atoms. In

general, this section has the feel of being less well developed than the first section, and raises more questions:

Scheme 8. Scheme 8b, showing inverted polarization is a surprising result, which flies in the face of chemical intuition, and also the classic behavior of nucleophile addition to difluoroalkenes. Can this be supported using a computational analysis of the HOMO and LUMO?

Reply: The inverted polarization was concluded from our experimental regioselectivity and supported by calculation results. According to the reviewer's suggestion, we further analyzed the HOMO and LUMO (please see Figure I). The results showed that the LUMO is mainly located at the C1 site, which suggests the higher electrophilicity of the C1 site than C2. This is consistent with our experiment observation. In addition, we also calculated the NPA charge for these two sites (Figure II). The results gave more straightforward evidence that the charges are +0.811 and -0.194 for the C1 and C2 sites, respectively. Replacement of the two F atoms with two Cl atoms resulted in a normal polarization (-0.133 and -0.051). We have added the NPA results to the text (in Scheme 8).

Figure I. The LUMO analysis of the product 3c.

Figure II. The NPA charges on carbon atoms.

The mechanistic discussion of the double dehydrofluorination is less well developed than the removal of 1 F.

Scheme 9... the energy profile implies B can be isolated. Has this been attempted? Isolation of B, perhaps by deliberately introducing a difluoroalkene starting material and 1 would lend experimental evidence to the proposed mechanism, which relies on the surprising inverted polarization above, and this should be attempted.

A 31.1 kcal/mol ΔG from B to TS5 is calculated. This seems a little high for the reaction proceeding at 80 degrees in the time given. Again this is difficult to reconcile with the experimental data.

Reply: Thanks for your suggestion. Actually, we have tried to isolate the intermediate B, but failed. To rationalize this, we re-calculated the energy barriers of HDF processes in acetonitrile (the previous DFT calculations were performed with toluene as the solvent). The results were given in Figure III and IV. As shown in Figure III, in the second HDF

process, the barrier for the beta-F elimination is 21.9 kcal/mol, which is lower than that for the initial hydride transfer (24.0 kcal/mol). Hence, the reaction conditions suitable for the formation of the intermediate B also facilitate its subsequent beta-F elimination. The intermediate B cannot be isolated.

Figure III. Energy profiles for HDF of **3c** by **1** calculated at the (SMD)-M06-2X/6-311++G(2df,2p)//(SMD)-M06-2X/6-31+G(d) level of theory (in kcal mol⁻¹).

The situation is different for the first HDF (Figure IV), where the barrier for the beta-F elimination (20.7 kcal/mol) is roughly 2 kcal/mol higher than that for the initial hydride transfer (18.9 kcal/mol). This allows the intermediate A to be intercepted during the HDF process at room temperature, which is consistent with the fact of the experiment isolation of A. These calculation results were updated in the revised manuscript.

Figure IV. Energy profiles for mono-HDF of **2c** by **1** calculated at the (SMD)-M06-2X/6-311++G(2df,2p)//(SMD)-M06-2X/6-31+G(d) level of theory (in kcal mol⁻¹).

Scheme 10.... Is the steric demand of F high enough to account for the observed selectivity? Were dipole minimization effects considered? I am a little suspicious, since

two of the largest aryl groups screened, 4q and 4r have some of the poorest selectivity shown in this reaction. As I mention later, I am not convinced by the NMR data provided that the minor products are the other isomer. If the reaction is operative in other solvents, would different E/Z ratios be obtained?

Reply: Thank you for your questions. We are not sure whether dipole minimization effects play a role here. We isolated the two isomers of 4p and 4r and their NMR spectra were given in the revised SI. NMR and ESI-HR data confirmed their structures. The 1D NOE NMR experiments of 4p was added to show their configuration information. And the similar analysis was also applicable to the analog compounds 4q and 4r. We tried to identify the E/Z ratios in toluene and THF; however, only the mono-HDF products were obtained in low yields. The low reactivity in these two solvents prohibited the second HDF.

Scheme 11... the structure of 4 is drawn incorrectly. The bonds are crossed, and the Z rather than E isomer is depicted.

Reply: Thanks. This mistake has been corrected.

Line 204 – The synthesis of 1 should also reference Gudat's pioneering work.

Reply: Thanks! The references were cited as refs. 73 and 74.

The supporting information appears to be lacking several key pieces of data required to support the characterization and assignment of the observed isomer ratios of the double defluorination products.

Line 82. While a reference was included, it would be helpful for the readership to have a scheme showing the procedure.

Reply: Thank you for your suggestion. We added a scheme to SI to show the processes for synthesis of trifluoromethyl alkenes.

Line 43: The boiling point of acetonitrile is 82 degrees. The temperatures reported are of the block or bath. Was it ascertained by internal thermocouple that the reaction temperature equilibrated to that of the block or bath?

Reply: Yes, we ascertained that. When the block temperatures were set at 60, 70, and 80 degrees, the temperatures of reaction solutions were about 59, 70, and 79 degrees, respectively. The temperature uncertainty is below a tolerable level.

Figure S1: I am surprised that 3J PF coupling is not observed in intermediate A. Can the authors comment? In addition, an expansion of the signal for 3a in acetonitrile (which has 2 F atoms in separate environments could be warranted).

Reply: Thanks. The 3J PF coupling in the intermediate A was observed in both the ^{19}F and ^{31}P NMR spectra with ^1H decoupling (Figure S1 and S2). In the ^{19}F NMR spectrum, the intermediate A shows a doublet (-62.29 and -62.32 ppm), corresponding to the quartet (99.45, 99.36, 99.27 and 99.19 ppm) in the ^{31}P NMR spectrum. To make them more visible, we have expanded these peaks (as well as those for 3a) in Figure S1-S3 in SI.

Spectra S1 and S2 are not integrated. Is quantitative integration with these nuclei possible?

The ratio of A to 1-F in figure S1 seems much larger than the ratio of A to 1F in figure S2. Figure S4. This figure appears to be a ^{31}P spectrum, but is labelled as a ^{19}F spectrum. Can the authors assign the broad peak at 190 ppm? This seems consistent with 1-Br, or another halogen other than F, which does not seem consistent with the reaction mixture.

Reply: Thanks for your concern and the questions. Because the intermediate A has 3 fluorine atoms, the area ratio of A to 1-F in the ^{19}F NMR spectrum (Figure S1) is 3 times of that in the ^{31}P NMR spectrum (Figure S2). The spectra S1 and S2 are integrated, giving ratios of 3:1. The broad peak at 190 ppm (Figure S4) may be assigned to the decomposition products of 1-F, which may correspond to the minor undefined peaks in the ^{19}F NMR spectrum (Figure S3). Considering the fact that stoichiometric 1-F was produced and the reaction was performed at relatively high temperature (70 degree), a small amount of 1-F may decompose.

Line 200... should read "due to the low boiling point of the products". It should be specified what pressure was used, rather than "not to high"

Reply: Thanks! For the liquid products of low-boiling point, the solvent was evaporated by rotary evaporators at 450 mbar. We specified this pressure in the experimental part in SI.

I don't think the 99% yields are reasonable for the reaction scale/ multistep purifications. No MS or elemental analysis data is reported. Many of the reported NMR spectra have significant impurities. Several of the product compounds listed on page S10 and S11 are new, and should have at least HRMS data in addition to the NMR data reported. Compound 6 (line 312, page 11) does have such data.

Line 313 onward concerns the monofluoroalkenes, formed by removal of 2 F atoms.

The yields are not at all realistic given the low NMR purities of these compounds. In addition, the integrals used to determine the reported isomer ratios are not shown on the images of the spectra.

Reply: Thank you for your good questions. We added the ESI-HR data for all the new compounds and integrated all the NMR peaks. Some of HDF experiments (including substrates 4g, 4k, 4p, 4q and 4z) were re-performed to identify the reaction yields. The NMR spectra of the reaction mixture showed complete conversion of substrates to the desired products. And the isolated yields are consistent well with our previous results. The mentioned NMR spectra with significant impurities were re-acquired and updated in SI. Furthermore, we carried out gram-scale synthesis of products **3j**, **3i**, and **4i**, all gave excellent isolated yields (91-96%). Some notable impurities in NMR spectra are assigned to the residual solvent and incompletely deuterated CDCl_3 . The integral areas for the actual impurities in NMR spectra are generally small (usually below 1%). For details, please see the revised SI.

Line 317- Complete dehydrofluorination is a bit confusing, since one F remains.... Perhaps "conversion to desired product"

Reply: Thanks for your suggestion. Accordingly, we updated this term in SI.

Line 393- tabulated data for compound 4q is missing.

Reply: Thanks! These data were provided in the revised SI.

Line 401-data for Z isomer in ^{19}F NMR is not shown in tabulated form. I have my doubts about the assignment of the minor isomer in this case, given the dramatically different appearance of the minor signals in the NMR spectra (addressed later)

Reply: Thanks. We presented the data for Z isomer in tabulated form.

Line 608- Why is there such a dramatic shift difference in the ^{19}F spectrum between the "isomers". Are these in fact isomers?

Reply: Thanks! The peaks at about -87 ppm is the impurity. We have purified the product and updated the NMR spectra of product 4r in the revised SI.

These comments address the images of the NMR spectra:

Line 414- Integrals are missing from ^1H NMR spectrum.

Line 424- The ^1H NMR spectrum contains aliphatic impurities, and integrals are not shown.

Line 428- Integrals are missing from ^1H NMR spectrum.

Line 433 Integrals are missing from ^1H NMR spectrum.

Line 448 Integrals are missing from ^1H NMR spectrum.

Line 461- Integrals are missing from ^1H NMR spectrum.

Reply: Thanks you for your comments. We added the integrals for all the peaks in NMR spectra, accordingly.

A number of the product NMRs have extra peaks

For example line 506 ^{19}F spectrum for 3i

Double dehydrofluorination products:

Line 565 onward. The products are messier. How was the configuration determined? The integrals that were used to select the geometric ratios were not shown.

Products such as 4g (line 574) are quite impure.

Reply: Thank you for all your good comments and suggestions! These products were purified and corresponding yields were re-calculated. We integrated all the peaks in NMR spectra, including those for the determination of the geometric ratios. Please see the revised SI for details.

Reviewer #2 (Remarks to the Author):

In the manuscript, a novel process involving trifluoromethyl-substituted styrenes is described. The reaction proceeds via new type of mechanistic cycle base on a catalyst bearing the phosphorus-hydrogen bond. Importantly, no transition metal catalyst is needed. A key feature of the manuscript is the addition of the P-H bond across the C,C double bond. The proposed mechanistic cycle is supported by quantum chemical calculations. It should be noted that this reaction mechanism stands in sharp contrast to many typical reactions of trifluoromethyl-substituted styrenes. The results are nicely presented. Overall, given the novelty of the results, I would suggest acceptance of the manuscript.

Reply: Thanks very much for your nice words and recommendation!

Additional comments.

1. For all solid products, melting points must be given (even for known compounds).

Reply: Thanks! We added the melting points for all solid products to SI.

2. For new compounds, either HRMS or combustion analysis must be given (e.g. 3f, 3q, 3r, and many others).

Reply: Thank you for your suggestion We gave the ESI-HR data for all the new products. For details, please see the revised SI.

3. In the procedures: "...and concentrated under vacuum (Note: due to the low boil of products, the vacuum should not be too high)" This would be difficult to reproduce. Some hint about the vacuum would be helpful.

Reply: Thanks for your comment. For the liquid products of low-boiling point, the solvent was evaporated by rotary evaporators at 450 mbar. We specified this pressure in the experimental part in the revised SI.

Reviewer #3 (Remarks to the Author):

This manuscript by Yang, Cheng and co-workers presents a very important work regarding the metal-free catalytic hydrodefluorination of trifluoromethylalkenes.

The authors first examined the stoichiometric reaction of diazaphospholene [1], with α -trifluoromethyl-styrene [2a]. This gives gem-difluoroalkene [3a] and fluorinated diazaphospholene [1-F], concomitant with the hydrophosphination intermediate [A]. Subsequently, the authors confirmed the formation of [1-F] from [2a] by heating the reaction mixture. This discovery is very important. In the reported diazaphospholene chemistry, a beta-elimination process has never been achieved, which hampers the development of transition metals-like catalytic reactions. By employing PhSiH₃ as a reducing reagent, reproduction of [1] from [1-F] is successfully achieved, which allows establishing the catalytic hydrodefluorination cycle. Moreover, by simply adjusting the reactant stoichiometry, gem-difluoroalkenes and terminal monofluoroalkenes are selectively obtained. The scope of substrates is nicely examined, the transformation of Indometacin derivatives has successfully been achieved with the authors' system, and the computational studies provide the reasonable reaction pathways, in good agreement with the experimental observation.

Overall, the study is well done both experimentally and theoretically, I do not see any major technical issues. The discovery of the beta-elimination process involved in this catalytic reaction is very significant and highly original. The present work should get a wide readership and I recommend the acceptance for publication after the very minor revision on the points indicated below:

Reply: Thank you very much for your positive comments and recommendation!

(i) While the proposed mechanism sounds reasonable, have the authors considered a process involving radical species? Have the authors tried the reaction under the light-shielded conditions?

Reply: Thanks for your suggestion. We found that the light had no impact on our results.

(ii) By increasing the amount of PhSiH₃, is it possible to complete HDF of compound 4?

Reply: When the amount of PhSiH₃ was increased, compounds 4 could be further defluorinated, but, in quite low yields.

(iii) When F atoms are replaced with other elements such as Cl or Br, do the similar reactions still proceed? Any comments on the importance of the F atoms from a viewpoint of the mechanism ~ energy would be highly appreciated.

Reply: Thank you for your question. There are several crucial factors for the success of the present system. 1) The formation of a stronger P-F bond facilitates the beta-F elimination. 2) In the first HDF, the strongly electron-withdrawing CF₃ group promotes the addition of the hydride to the alkene terminal carbon (Fig. Va), which allows the P moiety to connect to the carbon adjacent to CF₃ (the purple dot). 3) Thanks to the unexpected electron-donating effect of alkene terminal fluorine atoms, the second addition of the P-H bond to alkenes can also adopt an appropriate orientation to ensure the subsequent P-F elimination (Fig. Vb). 4) The first dehydrochlorination is feasible based on the calculated activation barrier if F atoms are replaced with Cl atoms (Fig. Vc). However, the second hydridic reduction would give an inverted regioselectivity and the subsequent P-Cl elimination cannot occur (Fig. Vd). This inverted polarity was supported by the NPA analysis (Fig. VI).

Figure V. Reaction pathways for HDF.

Figure VI. The NPA charges on carbon atoms.

(iv) Regarding the recent development of diazaphospholene catalysis, neither the recent

reviews (for instance, Chem. Soc. Rev., 2020, 49, 8335) nor original reports therein have been cited at all. Those should be cited properly.

Reply: Thank you for the good suggestion. This work was cited as ref. 61.

REVIEWERS' COMMENTS

Reviewer #1 (Remarks to the Author):

I was Reviewer #1, and had extensive commentary. In short, I will say I am happy with the authors' efforts to address my concerns.

The authors have added several relevant citations. In addition they have made a number of helpful revisions to the text. This is based on quite a large experimental effort, and they should be commended for the short turnaround time. They have nicely addressed/ rebutted several points I raised about the mechanism of the second HDF, and I am satisfied by their responses.

I would suggest a few small changes. On page 7, I would suggest changing the definitive "cannot" to the more accurate "did not".

In Scheme 10, the Newman projections shown leading to elimination are shown in a staggered conformation, but would an eclipsed conformation not be more accurate? In essence, I have to rotate in my head to see what will lead to Z vs E.

Reviewer #2 (Remarks to the Author):

Authors responded to all my comments, and made necessary corrections. The revised manuscript can be accepted for publication.

Reviewer #3 (Remarks to the Author):

I think that the authors have addressed nicely to the previous reviewers' concerns, and provide reasonable and satisfactory responses.

I think the present version is nearly ready for acceptance. Prior to the final decision, the following very minor changes are required.

(i) In the introduction, the authors state "This first metal-free hydrodefluorination ~". This description sounds inaccurate/overstatement as there are several reports on 'metal-free hydrodefluorination'. For example, see: Science (10.1126/science.abg0781 (2021)), JACS (2020, 142, 2572, 10.1021/jacs.9b12167), JACS (2010, 132, 4964, 10.1021/ja100605m), ChemEurJ (2017, 23, 17692, 10.1002/chem.201705276), Science (2008, 321, 1188, 10.1126/science.1159979), etc..

(ii) There are two exactly same references: ref 65 and ref 67.

(iii) References 60, 62, 63, 65 67 are missing the journal name.

Point-by-point Responses to the Reviewers' Comments (NCOMMS-21-04261A)

Reviewer #1 (Remarks to the Author):

I was Reviewer #1, and had extensive commentary. In short, I will say I am happy with the authors' efforts to address my concerns.

The authors have added several relevant citations. In addition they have made a number of helpful revisions to the text. This is based on quite a large experimental effort, and they should be commended for the short turnaround time. They have nicely addressed/ rebutted several points I raised about the mechanism of the second HDF, and I am satisfied by their responses.

Reply: Thanks very much for your nice words and recommendation.

I would suggest a few small changes. On page 7, I would suggest changing the definitive "cannot" to the more accurate "did not".

Reply: Thanks for your suggestion. Accordingly, we revised "cannot" into "did not".

In Scheme 10, the Newman projections shown leading to elimination are shown in a staggered conformation, but would an eclipsed conformation not be more accurate? In essence, I have to rotate in my head to see what will lead to Z vs E.

Reply: Based on your suggestion, we re-drew the Newman projections in an eclipsed conformation.

Reviewer #2 (Remarks to the Author):

Authors responded to all my comments, and made necessary corrections. The revised manuscript can be accepted for publication.

Reply: Thanks very much for your recommendation.

Reviewer #3 (Remarks to the Author):

I think that the authors have addressed nicely to the previous reviewers' concerns, and provide reasonable and satisfactory responses.

I think the present version is nearly ready for acceptance. Prior to the final decision, the following very minor changes are required.

Reply: Thanks very much for your recommendation.

(i) In the introduction, the authors state "This first metal-free hydrodefluorination ~". This description sounds inaccurate/overstatement as there are several reports on 'metal-free hydrodefluorination'. For example, see: Science (10.1126/science.abg0781 (2021)), JACS (2020, 142, 2572, 10.1021/jacs.9b12167), JACS (2010, 132, 4964, 10.1021/ja100605m), ChemEurJ (2017, 23, 17692, 10.1002/chem.201705276), Science (2008, 321, 1188,

10.1126/science.1159979), etc..

Reply: We are sorry for causing your confusion. Actually, the present work is the first chemoselective hydrodefluorination of trifluoromethylalkenes under metal-free conditions, considering that our manuscript was submitted to *Nature Communications* prior to the recent acceptance of Science (10.1126/science.abg0781 (2021)). In order to clear up the unnecessary misunderstanding, we deleted the term "The first".

(ii) There are two exactly same references: ref 65 and ref 67.

(iii) References 60, 62, 63, 65 67 are missing the journal name.

Reply: Thanks. These terms were checked.